# Emerging Role of Balloon Pulmonary Angioplasty in Chronic Thromboembolic Pulmonary Disease—Insights from the 2022 ESC Guidelines

**DOI:** 10.3390/jcm12165336

**Published:** 2023-08-16

**Authors:** Marta Banaszkiewicz, Paweł Kurzyna, Nina Kubikowska, Magda Mucha, Aleksander Rudnik, Aleksandra Gąsecka, Arkadiusz Pietrasik, Marcin Grabowski, Miłosz J. Jaguszewski, Piotr Kasprzyk, Piotr Kędzierski, Dariusz Ciećwierz, Grzegorz Żuk, Piotr Szwed, Michał Piłka, Michał Florczyk, Marcin Kurzyna, Szymon Darocha

**Affiliations:** 1Chair and Department of Pulmonary Circulation, Thromboembolic Diseases and Cardiology, Center of Postgraduate Medical Education, European Health Center, ERN-LUNG Member, 05-400 Otwock, Poland; 21st Chair and Department of Cardiology, Medical University of Warsaw, 02-091 Warszawa, Poland; 31st Department of Cardiology, Medical University of Gdansk, 80-210 Gdansk, Poland

**Keywords:** chronic thromboembolic pulmonary disease, chronic thromboembolic pulmonary hypertension, balloon pulmonary angioplasty, pulmonary endarterectomy, pulmonary hypertension

## Abstract

In this article, we discuss the topic of chronic thromboembolic pulmonary disease (CTEPD) and the growing role of balloon pulmonary angioplasty (BPA) in its treatment. We present the pathophysiology of CTEPD which arises from an incomplete resolution of thrombi in the pulmonary arteries and leads to stenosis and occlusion of the vessels. The article focuses mainly on the chronic thromboembolic pulmonary hypertension (CTEPH) subpopulation for which prognosis is very poor when left untreated. We describe a multimodal approach to treating CTEPH, including pulmonary endarterectomy (PEA), BPA, and pharmacological therapies. Additionally, the benefits of pharmacological pre-treatment before BPA and the technical aspects of the procedure itself are outlined. It is emphasized that BPA does not replace PEA but serves as a complementary treatment option for eligible patients. We summarized efficacy and treatment goals including an improvement in functional and biochemical parameters before and after BPA. Patients who received pre-treatment with riociguat prior to BPA exhibited a notable reduction in the occurrence of less severe complications. However, elderly patients are still perceived as an especially vulnerable group. It is shown that the prognosis of patients undergoing BPA is similar to PEA in the first years after the procedure but the long-term prognosis of BPA still remains unclear. The 2022 ESC/ERS guidelines highlight the significant role of BPA in the multimodal treatment of CTEPH, emphasizing its effectiveness and recommending its consideration as a therapeutic option for patients with CTEPD, both with and without pulmonary hypertension. This review summarizes the available evidence for BPA, patient selection, procedural details, and prognosis and discusses the potential future role of BPA in the management of CTEPH.

## 1. Introduction

Chronic thromboembolic pulmonary hypertension (CTEPH) constitutes a separate group of the clinical classification of pulmonary hypertension (PH) presented in the recent Guidelines of the European Society of Cardiology and European Respiratory Society [1]. The newly proposed CTEPH treatment algorithm introduces a multimodal treatment approach which is a combination of pulmonary endarterectomy (PEA), balloon pulmonary angioplasty (BPA), and medical therapy. The model is to target mixed anatomical lesions: proximal, distal, and microvasculopathy, respectively [1]. PEA remains a treatment of choice for patients with occlusions limited to proximal and surgically accessible arteries whereas medical and interventional therapies have been successfully implemented for patients with inoperable disease [2]. However, indications for each therapy might overlap whereas the results of each treatment are not necessarily corresponding.

Given the increasing understanding of the role of BPA, with its effectiveness and safety in the selected group of patients, the ESC/ERS guidelines have upgraded their recommendation for BPA from IIb-C in 2015 to I-B in 2022 [1]. This paper discusses the emerging role of BPA in the treatment of chronic thromboembolic pulmonary disease (CTEPD) with or without PH within the context of the newly proposed treatment strategy.

## 2. Epidemiology and Pathophysiology of CTEPH

The incidence of CTEPH is estimated to be around 3% in the first two years after a symptomatic episode of acute pulmonary embolism [3]. However, up to 25% of patients with a CTEPH diagnosis have no history of a thromboembolic episode [4]. The pathophysiological basis of the disease lies in the incomplete resolution of thromboembolic lesions after at least three months of therapeutic anticoagulation. A characteristic feature is the fibrous remodeling of thrombi in the pulmonary arteries, resulting in permanent mechanical obstruction. It is unclear as to what determines incomplete thrombus resolution. Several factors are suggested, such as autoimmune diseases, hematological disorders, or multiple comorbidities that impair the lytic system [5]. However, PH in this setting may not only be a consequence of pulmonary vascular bed obstruction by organized fibrotic clots. It may also be associated with microvasculopathy as a result of an increased blood flow in thrombus-free arteries. The morphology of the changes in the damaged microvasculature is similar to that observed in pulmonary arterial hypertension (PAH). Altogether, the above processes result in an increase in pulmonary vascular resistance (PVR) which in turn leads to right heart failure (RHF) and death if left untreated [6].

## 3. History of BPA

The world’s first report on the use of BPA in the treatment of CTEPH dates back to 1988 from the Netherlands, presenting the case of a 30-year-old man who was ineligible for PEA and underwent three interventional procedures followed by a significant decrease in mean pulmonary artery pressure (mPAP) [7]. The first case series of BPA had not been reported until 2001 when it was revealed that despite a significant hemodynamic improvement there were some unfavorable data, such as a 61% incidence of periprocedural complications and 5.6% death rate [1]. In the following years, the BPA technique was refined by Japanese interventionists, resulting in a new standard for this procedure set by 2012 [8,9]. The results of the first multicenter registry of BPA were published in 2017 by a group of Japanese researchers and included 1408 procedures performed in 308 CTEPH patients [10]. To date, also largest European series, describing German [11], French [12] and Polish [13] experience have been published.

## 4. Patient Management and Selection

The decision on an appropriate treatment option for CTEPH patients should be made individually and discussed in high-volume centers by a multidisciplinary team involving PEA surgeons, BPA interventionists, PH physicians, and radiologists. In clinical practice, pre-operational imaging is usually performed using computed tomography pulmonary angiography (CTPA) and digital subtraction angiography (DSA) which is still considered as the gold standard (Figure 1).

However, 3D printing (3DP) and augmented reality (AR) could mirror challenging patients’ anatomy in relation to three-dimensional relationships between structures more accurately than conventional angiography. Thus, both 3DP and AR would be additional effective tools in navigation, planning, and guiding transcatheter pulmonary interventions, as reported by Witowski et al. [15] (Figure 2).

Additionally, electrocardiographic features of right ventricular hypertrophy (RVH) were analyzed in newly diagnosed CTEPH patients to differentiate between the distal and proximal clots localization. It was revealed that only 8 out of 23 ECG RVH criteria were useful for differentiating between proximal and distal CTEPH localization and RV1 and SV6 may contribute as potential discriminators [16].

There is great evidence that PEA in operable CTEPH patients can deliver a spectacular improvement in symptoms and leads to the normalization of pulmonary hemodynamics [4,17,18]. However, some patients cannot be operated on. BPA and medical therapy are therapeutic options in symptomatic patients with inoperable disease or in those suffering from PH persistent or recurrent after PEA. The criteria for inoperability chiefly involve the anatomical distribution of pulmonary arteries lesions but the final decision to undergo such a complicated procedure may also be determined by severe comorbidities and patient refusal. In the international prospective CTEPH registry performed between 2007 and 2009, 63% of CTEPH patients were considered operable and 57% (varies from 12% to 60.9%, center-specific) underwent surgery [4]. In contrast, Siennicka et al. analyzed patients’ subpopulations classified by CTEPH Team in the years 2015–2018 for various treatment options. In the study, only 32% of 160 patients were assessed as eligible for PEA [19]. Similar observations come from the largest Polish CTEPH registry. Of the 516 patients included in the registry, PEA was performed only in 120 (23.3%) patients [20]. The reason for this marked difference in the proportion of patients treated surgically and interventionally, in favor of BPA, is the buoyant development of the BPA program. However, it is worth noting that based on data from the French registry, the absolute number of PEA is not in decline as the introduction of less invasive treatment options has translated into a higher number of patients diagnosed with CTEPD with or without PH [21].

Previously, patients with proximal-type CTEPH and high perioperative risk were excluded from BPA which was associated with a significantly poor prognosis in this cohort. The procedure, despite even local efficacy, was associated with a significantly higher risk of serious complications when applied to proximal dilated arteries filled with organized fibrotic clots. Nevertheless, rapidly increasing experience in high-volume BPA centers has emerged with this method as a potential therapeutic option also for CTEPH patients with an important contribution of proximal lesions [22,23]. However, the ESC/ERS Guidelines highlighted that BPA is not a replacement for PEA but rather an advancing treatment for patients who are not eligible for surgery.

Regardless of the high effectiveness of PEA, residual PH after surgery was reported in 17–31% of patients [18,24]. Clinically significant persistent PH after PEA, associated with a notably worse prognosis, occurs when mPAP is above 38 mmHg [24]. To date, few single-center reports involving a small number of patients have been published, advocating BPA as a complementary therapeutic strategy for recurrent or residual PH after PEA [25,26,27].

Finally, ESC/ERS guidelines provide IIa-C recommendations on both PEA and BPA in the treatment of symptomatic patients suffering from CTEPD without PH [1]. To date, only a small series was published in this area [28] supporting this treatment option. Consequently, the recently published ESC Consensus on BPA in CTEPD recommended that both methods be considered in this selected group of symptomatic patients. The multiparametric approach consisting of cardiopulmonary exercise testing, echocardiography, lung function testing, BNP/NT-proBNP, and chest radiography is thought to be useful in the identification of these patients [29].

## 5. Procedural Details

BPA is delivered as a series of staged procedures. BPA, like PEA, is a causal treatment for CTEPH involving the mechanical restoration of blood flow through the pulmonary arteries. The essence of the intervention is to disrupt the obstructive material within the pulmonary arteries without physically removing organized thrombi [30].

### 5.1. Patient Preparation

While all patients with CTEPD with or without PH require lifelong anticoagulation, there are no clear recommendations for the management of anticoagulation treatment in the perioperative period. The management is therefore center-specific. In patients treated with vitamin K antagonists (VKA), the treatment is most often continued with a titrate to an INR of approximately 2.0 prior to BPA. However, some centers discontinue VKA and opt for a bridging therapy with low-molecular-weight heparin (LMWH). Similarly, in some patients treated with direct oral anticoagulants, anticoagulation is discontinued immediately prior to the BPA session and in others it is discontinued early enough with the bridging therapy. However, it is worth noting that bridging therapy is regarded to be linked with a higher risk of bleeding [31].

### 5.2. Pretreatment with Targeted Pharmacotherapy

The ESC/ERS Guidelines provide a class IIa recommendation for targeted medical therapy prior to BPA to manage the microvascular component of CTEPH [1]. Riociguat, the soluble guanylate cyclase (sGC) stimulator, has the highest class of recommendation but other pulmonary arterial targeted drugs, such as phosphodiesterase type 5 inhibitors (PDE5i), endothelin receptor antagonists (ERA), or parenteral prostacyclin analogs may also be considered. A summary of pulmonary arterial targeted drugs tested in the CTEPH patient population is presented in Table 1. Moreover, a combination of sGC stimulator/PDE5i, ERA, or pareneteral prostacyclin analogs may be used in selected groups of patients [32,33]. Consequently, oral combination therapy including PDE5i and ERAs is a common practice in CTEPH patients with severe hemodynamic compromise [34]. The excellent results of the multimodal therapy with riociguat and BPA has been demonstrated in the study of 36 consecutive patients with inoperable CTEPH treated with riociguat for at least three months before BPA. Significant improvements in pulmonary hemodynamics and physical capacity have been observed for targeted medical treatment and subsequent BPA procedures yielded further benefits [35]. Moreover, pretreatment with riociguat improves hemodynamics and has been reported to reduce BPA-related complications. Thus, patients with a PVR above four Wood units should be treated with riociguat before starting interventional treatment [36]. To date, there are no clear recommendations regarding the termination or continuation of treatment after completion of BPA. The decision to continue or discontinue pulmonary arterial targeted treatment should be made individually based on the patient’s symptoms, number of remaining lesions, hemodynamic parameters, and previous response to treatment [29].

### 5.3. Technical Aspects

As BPA is the complex procedure requiring a detailed pulmonary vascular anatomy, it should be performed in catheter laboratory with biplane cineangiography [29,47]. In the process of preprocedural planning, contrast allergy and renal and/or thyroid dysfunction should be taken into consideration and handled according to general guidelines recommendations [29]. The procedure is performed in patients under local anesthesia. Vascular access is achieved via the femoral or jugular vein. It is necessary to perform right heart catheterization (RHC) before each BPA session to assess a number of hemodynamic parameters. There are two anticoagulation strategies most commonly used during BPA. The first one is an intravenous bolus of 2000–5000 units of unfractionated heparin (UFH) at the beginning, then with 1000 units of UFH per every hour. The other one is full anticoagulation at an activated clotting time (ACT) of 250 s as practiced in coronary interventions. Oxygen needs to be administered to maintain a saturation level of more than 92% during the procedure. A 6F guiding catheter (e.g., Judkins, Multipurpose, or Amplatz shape) is inserted into the right or left pulmonary artery through a 90 cm 6F vascular sheath. Then, the 0.014-inch coronary guidewire is passed through the lesions and inserted distally in the subsegmental artery. Consequently, the target branches are dilated with multiple balloon inflations using semi-compliant balloon catheters. The balloon diameter and length are tailored to the type of lesion, the degree of stenosis in the pulmonary artery observed with angiography, and the severity of PH. In line with the revised BPA strategy, at the initial phase of treatment the undersized balloon catheters of diameter 2.0–2.5 mm are used. Consequently, 1:1 sizing of balloon catheters is performed at the optimization phase when the mPAP is below 35 mmHg, as proposed by the group from Japan [8] (Figure 3).

Based on the lesion distribution and angiographic characteristics, there are five types of thromboembolic lesions as follows: ring-like stenosis lesion, web lesion, subtotal lesion, total occlusion lesion, and tortuous lesion. Primarily, ring and web lesions are addressed with a preference for branches of the right lung basal segment whereas total occlusion lesions should be dealt with once pulmonary hemodynamics have improved [48]. The key factor for achieving a good hemodynamic effect is maximally extensive revascularization. Thus, angioplasty should be gradually pursued within all lung segments [49,50]. In general, according to ESC Consensus on BPA, the minimum hemodynamic goal of BPA is a final mPAP below 30 mmHg. However, it should be individualized, especially in patients with comorbidities [29]. Details of the revised BPA strategy are summarized in Table 2.

In special cases, when the lesion characteristics are ambiguous, intravascular imaging, such as intravascular ultrasound (IVUS) (Figure 4) and optical coherence tomography (OCT) [51] may be applied.

The measurement of the pressure gradient across the evaluated lesion is not routinely practiced but some centers successfully apply pressure-wire-guided angioplasty. Finally, in order to protect against restenosis of proximal lesions, sporadic implantation of a stent may be required to manage an elastic recoil phenomenon which makes the BPA of a large vessel ineffective [52].

## 6. Efficacy and Treatment Goals

BPA is a well-established treatment method that effectively improves pulmonary hemodynamics, thereby reducing right ventricular afterload and improving patient prognosis.

The first and largest multicenter BPA registry, including 249 patients at follow-up, was conducted in seven Japanese institutions and revealed an overall decrease in mPAP of 48% after BPA and a decrease in PVR of 66%. It was reported that mPAP reached near normal values (decrease from 43.2 ± 11.0 to 22.5 ± 5.4 mmHg, *p* < 0.001) whereas PVR declined from 10.7 ± 5.6 to 3.6 ± 2.4 WU (*p* < 0.001) [10]. In Europe, an 18–29% decrease in mPAP with around a 42–45% decrease in PVR was revealed after the implementation of the improved BPA technique [11,12,13,53,54,55]. In detail, in the French data set the final postprocedural mPAP was 31.6 mmHg and PVR was 4.1 ± 2.2 WU [12] whereas in the German registry it was 33.0 ± 11.0 mmHg and 5.5 ± 3.5 WU [11], respectively. The polish multicenter registry revealed a significant decrease in mPAP from 46.1 ± 10.6 to 32.7 ± 10.9 mmHg (*p* < 0.001) as well as in PVR from 7.9 ± 5.5 to 4.3 ± 2.2 Wood Units (*p* < 0.001) [13]. Altogether, the improvement in pulmonary hemodynamics in the European reports is not as remarkable as that reported by the Japanese interventionists. Interestingly, these more satisfactory results in Japan were achieved despite the lower median BPA procedures per patient: four in Japan [10], four and half in Poland [13], five in Germany [11]. These could be due to different patients’ characteristics, suggesting less advanced disease and fewer comorbidities within the Japanese population as well as to different inflammatory thrombotic phenotype of intravascular lesions [56]. The details of hemodynamic improvement after BPA in hitherto published data sets are presented in Table 3.

As BPA seemed to improve hemodynamic parameters to a similar degree as PEA, the exact comparison with PEA would be of great interest. The retrospective study conducted to evaluate the efficacy and safety of BPA in 29 patients with inoperable CTEPH using the results of PEA for 24 operable patients as a reference suggested similar efficacy and the safety of both procedures in their target cohorts. Patients who underwent BPA presented improvement in mPAP from 39.4 ± 6.9 mmHg to 21.3 ± 5.6 mmHg (*p* < 0.001) and for PVR from 9.54 to 3.55 WU (*p* < 0.001) while patients after PEA had similar effects with decreased mPAP (from 44.4 ± 11.0 mmHg to 21.6 ± 6.7 mmHg, *p* < 0.001) and reduced PVR (from 9.76 WU to 3.23 Wood units, *p* < 0.001) [64]. Ravnestad et al. also compared PEA and BPA in 96 consecutive patients. At the baseline, no significant between-group differences were observed regarding hemodynamics or exercise capacity. Both BPA and PEA significantly reduced mPAP (from 43 ± 12 mmHg to 31 ± 9 mmHg; *p* < 0.001 and from 46 ± 11 mmHg at baseline to 28 ± 13 mmHg at follow-up; *p* < 0.001) and PVR (from 544 ± 322 dyn s cm^−5^ to 338 ± 180 dyn s cm^−5^; *p* < 0.001 and from 686 ± 347 dyn s cm^−5^ to 281 ± 197 dyn s cm^−5^; *p* < 0.001), with significantly lower reductions for both parameters after PEA. There were no significant differences between BPA and PEA regarding improvements in exercise capacity [65].

The efficacy of BPA in comparison to riociguat was evaluated in the RACE study. This randomized control trial revealed a reduction in PVR to be more pronounced with BPA than with riociguat. Specifically, PVR decreased to 39.9% of the baseline PVR in the BPA group and to 66.7% of the baseline in the riociguat group. As riociguat and BPA target different lesions within the pulmonary vascular bed, they also act differently on pulmonary hemodynamic parameters. Hence, in the study, CO increases more with riociguat (+0.7 vs. +1.1 L/min/m^2^, *p* 0.033) whereas mPAP decreased more with BPA (−18.7 vs. −5.1 mmHg, *p* < 0.0001) [36]. These results are in line with those revealed in a meta-analysis comparing riociguat with BPA therapy [66]. Therefore, the overall BPA efficacy has to be assessed together with ongoing vasodilator therapy.

The improvement in pulmonary hemodynamics after BPA translates into an improvement in right ventricular function, functional class (FC), and quality of life [67,68]. Changes in functional and biochemical parameters before and after BPA in hitherto published data sets are summarized in Table 4. As the right ventricle dysfunction is closely related to the unfavorable prognosis in CTEPH [69], restoring RV remodeling and systolic dysfunction in the course of BPA treatment is clinically beneficial. The improvement in right ventricular function after BPA was confirmed by cardiac magnetic resonance imaging [70], echocardiographic assessment [67], and ECG [71].

Correspondingly, BPA seems to be effective not only in patients with distally located thromboembolic lesions but also in patients with proximal-type CTEPH. Darocha et al. recorded that when it comes to improvements in mPAP and PVR, they did not contradict within the groups with surgically accessible and inaccessible lesions. However, only 16 patients with proximal-type CTEPH were enrolled in the study [23]. Contrarily, Nisihara et al. had 344 patients enrolled in their study out of whom 81 patients were those with surgically accessible lesions; they reported percent differences in mPAP (−37.8% vs. −48.9%; *p* = 0.005), PVR (−51.6% vs. −60.8%; *p* = 0.006), and 6MWD (+13.5% vs. +28.9%; *p* = 0.017). Moreover, the achievement of WHO FC I (14% vs. 30%; *p* < 0.05) after BPA was lower in the surgically accessible group compared to the inaccessible group [22].

Few studies assessed the efficacy of BPA as a complementary treatment in patients with persistent or recurrent PH after PEA. Shimura et al. delivered a systematic analysis of nine patients with persistent PH after PEA. The study revealed that hemodynamics and symptoms after PEA were significantly improved with supplemental BPA. What is worth noting is that they observed a low rate of serious procedure-related complications [27]. Further reports in this field were provided by Araszkiewicz et al. The authors enrolled 15 patients with PH persistent after PEA and demonstrated improvement in hemodynamic parameters, such as a 31% decline in mPAP and 38% decline in PVR [26]. Similar satisfactory results were obtained a few years later by Andersen et al. However, the latter analysis indicates a higher risk of complications in the group of patients who had previously undergone PEA compared to the group without prior surgery [72]. On the contrary, a hybrid approach seems to be particularly successful in patients with highly increased PVR before PEA which substantially extends their perioperative risk. Wiedenroth et al. researched a small group selected for the hybrid procedure where BPA was implemented alongside PEA. The authors found out that combining interventional BPA and surgical PEA served as an efficient treatment method for carefully selected patients with a high risk of CTEPH [47].

In patients with CTEPD without PH, BPA can also produce favorable outcomes and improve an abnormal PVR response to exercise [28,73]. However, as mentioned above, these observations resulted only from studies on small cohorts. Furthermore, these retrospective studies were conducted before the new definition of PH was introduced and patients included in the study had mPAP in the range of 20–24 mmHg and PVR of about 2 WU at rest. Currently, according to the new definition, the majority of patients included in those studies would have been diagnosed with CTEPH. Hence, larger prospective clinical trials are needed to better evaluate BPA treatment in the population with CTEPD without PH.

## 7. Safety

The early mortality rate of BPA ranges from 0% to 14.3% whereas the incidence rates of lung injury, hemoptysis, and vascular perforation ranged from 7.0% to 31.4% and 5.6–19.6% and 0–8.0%, respectively [74]. In the Japanese registry, pulmonary injury (17.8%) and hemoptysis (14.0%) were registered as the most common complications whereas pulmonary artery perforation, dissection, and rupture constituted only 2.9%, 0.4%, and 0.1%, respectively. On the other hand, in the European registries fewer complications were reported, with a lung injury rate of 9.4%, 9.1%, and 6.8% in German, French, and Polish data sets, respectively [10,11,12,13]. The lower complication rate in Europe may be explained by the fact that Japanese interventionists enrolled patients earlier than European institutions which were later able to benefit from the increasing BPA experience. Otherwise, the higher complication rate may be the result of more aggressive interventions performed by Japanese interventionists which could also support their greater hemodynamic effectiveness.

As BPA complication rates are greatly influenced by the definition, they should be clearly defined and uniformly reported. Thus, a guide for BPA centers to clearly classify complications has already been proposed [75]. Formerly, lung injury (LI) was regarded as a main and severe complication in the early stage [2,8,9]. The incidence of LI was reported to be 61% in the original BPA case series performed by Feinstein et al. [1] and varies from 30 to 60% in some refined BPA studies [9,76,77,78] though the definition has differed among studies. The LI is possibly caused by wire injury, balloon over dilatation, or high-pressure injection of contrast medium and is defined by new ground-glass opacities, consolidation, and pleural effusion in the area of treated arteries, assessed by CT within 24 h after BPA (Figure 5).

It may be accompanied by hemoptysis and/or hypoxia [9,10,76,77]. There are some preliminary reports that the occurrence of LI may be associated with a significant increase in plasma biomarkers, e.g., sST2 as early as 24 h after the procedure but the underlying mechanism remains unclear [79]. According to the Inami classification there are different degrees of LI severity: mild (defined as a need for supplemental oxygen via pongs), moderate (defined as a need for high-flow oxygen via a mask), and severe (requiring non-invasive positive pressure ventilation or mechanical ventilation and/or extra corporal mechanical oxygenation). The average mortality is 2% and differs from 0 to 10% depending on the degree of severity. Independent risk factors of LI are unfavorable hemodynamic parameters such as high mPAP before BPA [63,76]. In the RACE study discussed above, piecewise logistic regression identified mPAP above 45 mmHg as a predictive factor related to BPA adverse events. Furthermore, a less severe hemodynamic compromise was reported in the course of BPA in patients pretreated with riociguat compared to patients treated first with BPA [36]. Complication rates and the risk of death may also depend on the lesion characteristics, BPA technique, and operators. As reported in the French registry, complication rates fell from 11.2% per session in the first 1006 sessions to 7.7% in the most recent 562 procedures [12].

Non-specific complications, such as access-site problems, infections, or allergic reactions, are rarely seen and do not exceed 1.5% [13].

Elderly patients seem to be a particularly vulnerable group for a severe course of BPA complications. However, the reports of BPA safety in the elderly population are limited [80,81]. Nagai et al. evaluated the outcomes of BPA in CTEPH patients over 80 years of age. The safety and efficacy of the BPA procedures were compared in the elderly group (n = 14; ≥80 years; 66 sessions) and younger group (n = 59; <80 years; 278 sessions). The total number of complications, including mortality within 30 days of BPA, application of positive pressure ventilation, the occurrence of bloody sputum, and contrast-induced nephropathy after each session were similar between the two groups. Interestingly, a low ratio of occurrence of contrast-induced nephropathy was reported in both groups, particularly in the elderly group (0.4% in the younger group compared to 0% in the elderly group, *p* = 1.000) [80]. However, Darocha et al. in their study involving 41 patients demonstrated that BPA can be performed safely even in patients with chronic kidney disease [82]. Furthermore, renal function may be gradually improved as cardiac output increases as a result of interventional treatment [82,83].

Finally, there is limited evidence on the safety and efficacy of BPA treatment in very high-risk patients with comorbidities. Gościniak et al. presented a unique case of a 35-year-old patient with von Hippel-Lindau disease who developed CTEPH as a severe complication after a ventriculoatrial shunt for hydrocephalus and was successfully treated with six BPA procedures. [84]. Darocha et al. demonstrated a case of successful BPA treatment in a 46-year-old patient with a single lung after oncological surgery and heparin-induced thrombocytopenia. The patient completed a treatment of nine BPA sessions with no complications and sustained hemodynamic improvement as well as relief of dyspnea symptoms [85]. Most recently, the case of a successful rescue BPA procedure, performed in an 82-year-female with CTEPH and takotsubo cardiomyopathy, was published by Japanese researchers from Okayama Medical Center [86].

## 8. Long-Term Prognosis

As clots are not removed from the pulmonary arteries during BPA procedure, the long-term durability of hemodynamic effects remains uncertain. However, a recent study found that restenosis after BPA treatment is extremely rare and that the lumen diameter did not decrease compared to the post-BPA baseline [87]. Inami et al. in their paper reported on the long-term persistence of hemodynamic effects assessed in serial measurements for a minimum 3.5-year follow-up after the BPA series. Significant improvements in mPAP and PVR were maintained throughout the observation period while the initial improvement in cardiac index was not sustained at the long-term follow-up. That was explained by the high withdrawal rate from medical therapy during follow-up [88]. In 2017, Aoki et al. reported outcomes of 424 BPA sessions performed from 2009 to 2016. The 5-year survival rate of 98.4% without any periprocedural deaths was revealed [89]. At the same time, data from the Japanese registry showed overall 1-, 2-, and 3-year survival rates of 96.8%, 96.8%, and 94.5%, respectively [10]. Outside Japan, in the French experience the 1- and 3-survival rate was 97.3% and 95.1, respectively [12]. Most recently, in the Polish data sets, the overall survival rate was 92.4% for three years [13] and 88% for 5 years [23] after initial BPA procedure. Although the prognosis of patients undergoing BPA is comparable to those undergoing PEA surgery in the first years of follow-up, it is still unclear as to how BPA affects very long-term prognosis (>10 years) due to the small number of patients with sufficient follow-up periods.

## 9. Future Perspectives and Outstanding Issues

The introduction of BPA into the course of CTEPH treatment has been undoubtedly one of the greatest advances of the last decade in terms of the management of PH. The availability of several treatment modalities opens the possibility of using a combination therapy—either in a sequential manner or simultaneously—in selected groups of patients. Thus, a careful multidisciplinary assessment deliberating all possible treatment modalities (including combination strategies) for CTEPH has become more important than ever before. CTEPH registries to date have emphasized high variability in disease management depending on the level of experience of the center and the region [34]. The technique of the procedure has been excellently improved in the last 10 years. However, another key element requiring a clear consensus is the standardization of the BPA procedure and the therapeutic goals of the treatment provided. The latter issue is particularly relevant, especially in view of the scanty information on long-term survival as it is still unclear to what extent improving the hemodynamic parameters leads to improved outcomes. It also remains an open question as to whether targeted treatment should be continued after successful BPA treatment and whether PEA and BPA should be performed sequentially or simultaneously in patients with mixed pulmonary artery lesions. Furthermore, the subgroup of patients with symptomatic CTEPD without resting PH is a group that has not been adequately addressed in hitherto reports. The ongoing consecutive international BPA registry (NCT03245268) will play an important role in answering some of the aforementioned questions.

## 10. Conclusions

After many years of development and refinement, the BPA procedure has finally become a widely accepted and broadly used treatment option for patients with inoperable or residual PH. There is a promising body of evidence revealing the safety and efficacy of BPA which can greatly improve pulmonary hemodynamics, exercise capacity, and general quality of life. The 2022 ERS/ESC guidelines promote the development of new treatment strategies in which BPA, along with PEA and medical therapy, have a significant role for patients in whom a single treatment method failed to achieve satisfactory outcomes. Today’s CTEPH patient population consists primarily of elderly people with a significant burden of comorbidities. Thus, tailoring therapeutic decisions individually to each patient remains a key issue in achieving therapeutic success.

## Figures and Tables

**Figure 1 jcm-12-05336-f001:**
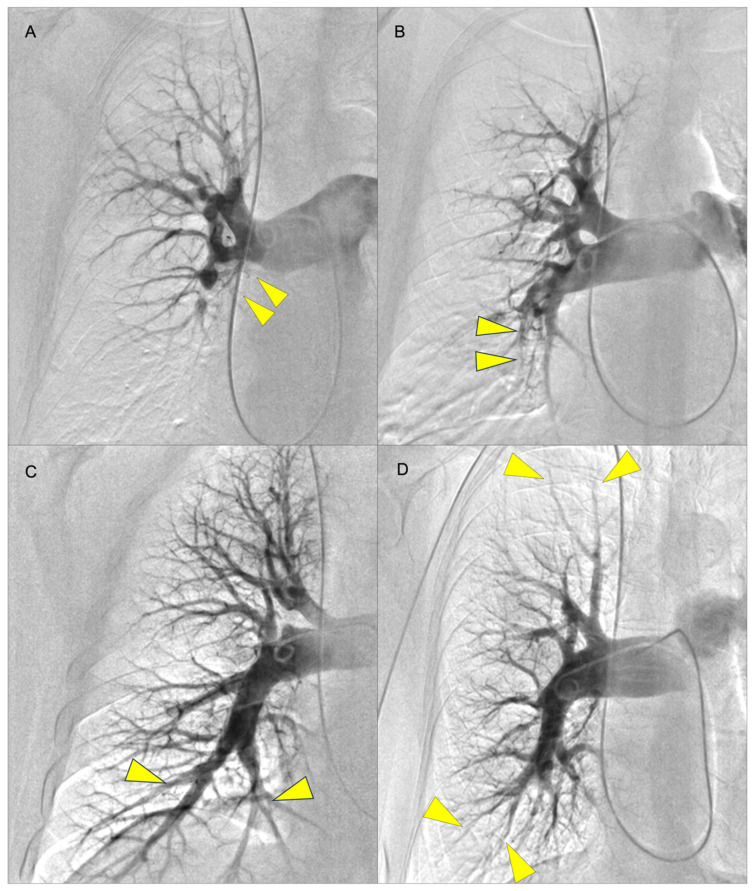
Examples of pulmonary angiography for levels I to IV according to the University of San Diego classification (yellow arrows indicate lesions): (**A**) level I (lesions starting in the main pulmonary artery with level IC corresponding to complete occlusion of one main PA); (**B**) level II (lesions starting at the level of lobar arteries or in the main descending PAs); (**C**) level III (lesions starting at the level of the segmental arteries); and (**D**) level IV (lesions starting at the level of the subsegmental arteries) [14]. The levels I–III are considered surgically accessible lesions whereas level IV is not operable.

**Figure 2 jcm-12-05336-f002:**
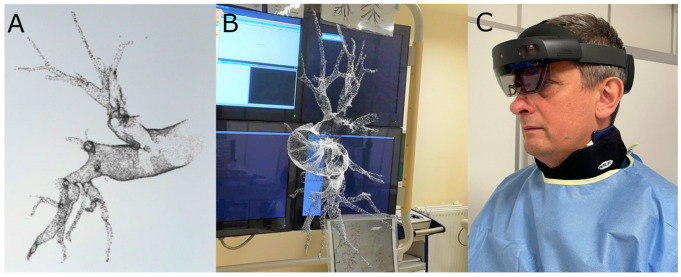
(**A**) Holographic reconstruction of a pulmonary artery created with CarnaLife Holo software (version 2.4.4.10379, Medapp SA, Kraków, Poland). (**B**) First-person view of the hologram of the pulmonary artery branches displayed throughout the BPA procedure; (Images courtesy of Grzegorz Bałda). (**C**) The operator using an AR headset (HoloLens 2, Microsoft) during one of the procedures.

**Figure 3 jcm-12-05336-f003:**
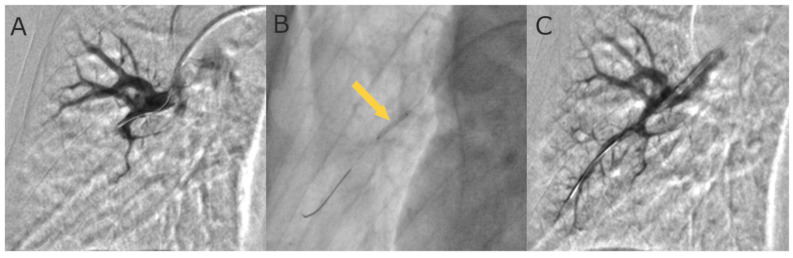
Selective pulmonary angiograms: (**A**) before BPA; (**B**) during 4.0 mm balloon inflation (arrow); and (**C**) after BPA.

**Figure 4 jcm-12-05336-f004:**
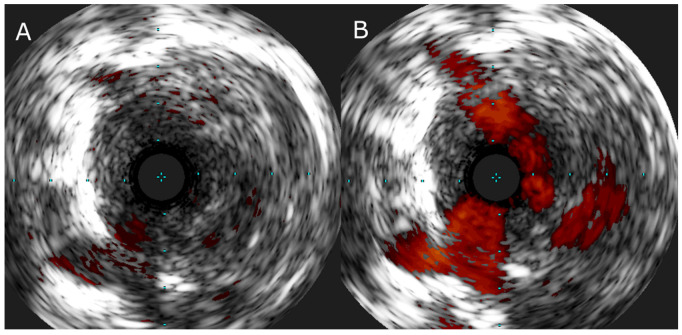
(**A**) IVUS image of the web lesion in a segmental branch of pulmonary artery and (**B**) IVUS image of the same lesion with color flow imaging showing recanalized flow channels in the central part of the vessel.

**Figure 5 jcm-12-05336-f005:**
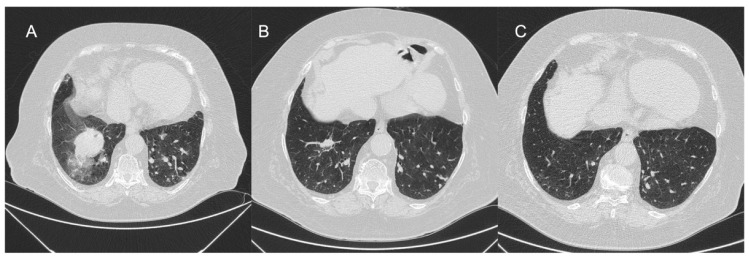
Representative CT image of lung injury. (**A**) Second day after BPA procedure; (**B**) 3 months after BPA procedure; and (**C**) 10 months after BPA procedure.

**Table 1 jcm-12-05336-t001:** The summary of the use of pulmonary arterial targeted drugs in chronic thromboembolic pulmonary hypertension.

Agent	First Author, Year	Study Type	Population [n]	Outcomes
Riociguat	Ghofrani [37] 2013	RCT riociguat vs. placebo	inoperable CTEPH or persistent CTEPH after PEA [174 riociguat, 88 placebo]	6mWT distance increased by 46 m PVR reduced by 246 dyn mPAP decreased by 5 mmHg NT-proBNP level decreased by 444 pg/mL
	Jaïs [36] 2022	RCT riociguat vs. BPA	inoperable CTEPH and PVR more than 320 dyn [53 riociguat, 52 BPA]	PVR reduction and WHO FC improvement was more pronounced in BPA group CI increased more in riociguat group
Sildenafil	Ghofrani [38] 2003	open-label pilot trial without control group	inoperable CTEPH and deterioration over 3 months [12 sildenafil]	PVR reduced by 574 dyn 6mWT distance increased by 54 m mPAP decreased by 7.7 mmHg
	Reichenberger [39] 2007	open-label trial without control group	inoperable CTEPH [104 sildenafil]	6mWT distance increased by 51 m PVR reduced by 104 dyn
	Suntharalingam [40] 2008	RCT sildenafil vs. placebo	inoperable CTEPH or persistent CTEPH after PEA [9 sildenafil, 10 placebo]	WHO FC improved in 44% cases more than one PVR decreased by 197 dyn
Bosentan	Hoeper [41] 2005	open-label pilot trial without control group	inoperable CTEPH or persistent CTEPH after PEA [19 bosentan]	PVR reduced by 313 dyn 6mWT distance increased by 73 m mPAP decreased by 6 mmHg NT-proBNP level decreased by 716 pg/mL
	Bonderman [42] 2005	case series	inoperable CTEPH [16 bosentan]	WHO FC improved by one in 69% cases 6mWT distance increased by 92 m NT-proBNP level decreased by 1610 pg/mL
	Jaïs [43] 2008	RCT bosentan vs. placebo	inoperable CTEPH or persistent CTEPH after PEA [77 bosentan, 80 placebo]	PVR decreased by 193 dyn no significant differences in WHO FC
Macitentan	Ghofrani [33] 2017	RCT macitentan vs. placebo	inoperable CTEPH [40 macitentan, 40 placebo]	PVR decreased by 16% NT-proBNP level decreased by 1610 pg/mL 6mWT distance increased by 34 m
Treprostinil (s.c.)	Skoro-Sajer [44] 2007	prospective uncontrolled observational cohort study	inoperable CTEPH or persistent CTEPH after PEA [25 treprostinil]	6mWT distance increased by 105 m WHO FC improved in 52% cases PVR decreased by 116 dyn
	Sadushi-Kolici [32] 2019	RCT high-dose vs. low-dose treprostinil	inoperable CTEPH or persistent CTEPH after PEA [45 high-dose, 46 low-dose]	6mWT distance increased by 53 m WHO FC improved in 51% cases NT-proBNP level decreased by 157 pg/mL PVR decreased by 214 dyn
Epoprostenol (i.v.)	Scelsi [45] 2004	retrospective study	inoperable CTEPH [11 epoprostenol]	WHO FC improved in 55% cases 6mWT distance increased by 99 m clinical signs of right heart failure resolute in 45% cases
	Cabrol [46] 2007	retrospective study	inoperable CTEPH or persistent CTEPH after PEA [27 epoprostenol]	WHO FC improved in 39% cases 6mWT distance increased by 66 m mPAP decreased by 4 mmHg

CI—cardiac index, CTEPH—chronic thrombo-embolic pulmonary hypertension, mPAP—mean pulmonary artery pressure, NT-proBNP—N-terminal prohormone of brain natriuretic peptide, PEA—pulmonary endarterectomy, PVR—pulmonary vascular resistance, RCT—randomized controlled trial, WHO FC—World Health Organization functional class, 6mWT—six minutes walking test.

**Table 2 jcm-12-05336-t002:** The initial stage and optimization stage of the revised strategy of balloon pulmonary angioplasty (BPA).

Initial Stage—“Air Strike”	Optimisation Stage
Use of 2.5 mm dimension balloon catheter	Balloon catheter diameter adjustment 1:1 to vessels using IVUS
Only one inflation for one lesion, without optimisation	Multiple inflations from distal to proximal segment of the vessels, optimising hemodynamic effects
As a priority: BPA in the vessels of the lower lobe and maximum available lesions in one lung	Staged treatment of all lesions; in any location in both lungs
No treatment for type C (subtotal occlusion) and D (total occlusion) lesions	Attempt of treatment type C (subtotal occlusion) and D lesions (total occlusion)
As standard: 2 BPA sessions, with a treatment target of <40 mmHg mPAP followed by an optimisation stage	About four to five sessions until treatment goals are achieved

BPA—balloon pulmonary angioplasty; IVUS—intravascular ultrasound; mPAP—mean pulmonary artery pressure.

**Table 3 jcm-12-05336-t003:** Improvement in hemodynamic parameters after balloon pulmonary angioplasty. Data are presented as mean ± SD unless stated otherwise.

First Author, Year	Patients [n]	Mean PAP [mmHg]			PVR [Wood Units]		
		Pre-BPA	Post-BPA	*p*-Value	Pre-BPA	Post-BPA	*p*-Value
Ogawa, 2017 [10]	308	43.2 ± 11.0	24.3 ± 6.4	<0.001	10.67 ± 5.63	4.49 ± 2.78	<0.001
Ogo, 2017 [57]	80	42 ± 11	25 ± 6	<0.01	11 ± 5.3	5.1 ± 2.3	<0.01
Olsson, 2017 [11]	56	40 ± 12	33 ± 11	<0.001	7.39 ± 3.575	5.5 ± 3.49	<0.001
Brenot, 2019 [12]	154	43.9 ± 9.5	31.6 ± 9.0	<0.001	7.55 ± 2.82	4.11 ± 2.21	<0.001
Velázquez, 2019 [58]	43	49.5 ± 12	37.8 ± 9	<0.001	10.1 ± 4.9	5.6 ± 2.2	<0.001
van Thor, 2020 [59]	38	39.5 ± 11.6	30.6 ± 8.2	<0.001	6.1 ± 4.7	3.3 ± 2.0	<0.001
Hoole, 2020 [60]	30	44.7 ± 11.0	34.4 ± 8.3	<0.001	8.29 ± 3.5	5.45 ± 2.45	<0.001
Jansa, 2020 [53]	25	39.7 ± 11.1	−7.4 ± 6.9 (absolute diff.) ^1^	<0.001	6.6 ± 3.0	−2.2 ± 1.6 (absolute diff.) ^1^	<0.001
Gerges, 2021 [61]	45	38.8 ± 10.7	25.5 ± 5.2	<0.001	6.2 ± 2.9	3.5 ± 1.7	<0.001
Darocha, 2021 [23]	70	48.6 ± 10	31.3 ± 8.6	<0.001	8.675 ± 3.7	4.1625 ± 2.025	<0.001
Karyofyllis, 2022 [62]	15	47.8 ± 13.5	26.4 ± 7.6	<0.001	10.0 ± 5.0	3.7 ± 2.8	<0.001
Wiedenroth, 2022 [63]	142	39.7 ± 11.2	28 (23–35) (median (IQR))	<0.001	6.6 (4.1–8.7) (median (IQR))	3.6 (2.6–5) (median (IQR))	<0.001
Darocha, 2022 [13]	156	45.1 ± 10.7	30.1 ± 10.2	<0.001	8.025 ± 4.26	4.05 ± 2.31	<0.001

absolute diff.: absolute difference; mPAP: mean pulmonary arterial pressure; NR: not reported; PVR: pulmonary vascular resistance; NT-proBNP: N-terminal fragment of pro-brain natriuretic peptide; RAP: right arterial pressure; SvO2: mixed venous oxygen saturation; CI: cardiac index. ^1^ Absolute difference was calculated as the difference between values from the last BPA and before the first BPA.

**Table 4 jcm-12-05336-t004:** Improvement in functional and biochemical parameters after BPA in hitherto published reports. Data are presented as mean ± SD or median (IQR) unless stated otherwise.

First Author, Year	Patients [n]	6MWD [m]			NT-proBNP [pg/mL]			NYHA-FC(I-II)	
		Pre-BPA	Post-BPA	*p*-Value	Pre-BPA	Post-BPA	*p*-Value	Pre-BPA	Post-BPA
Ogawa, 2017 [10]	308	318 ± 122	401 ± 104	<0.001	239 ± 334	43 ± 76	<0.001	3 (median)	2 (median)
Ogo, 2017 [57]	80	372 ± 124	470 ± 99	<0.01	227 ± 282	48 ± 57	<0.01	NR	NR
Olsson, 2017 [11]	56	358 ± 108	391 ± 108	0.001	504 (233–1676)	242 (109–555)	0.002	16%	88%
Brenot, 2019 [12]	154	396 ± 120	441 ± 104	<0.001	NR	NR	-	NR	NR
Velázquez, 2019 [58]	43	394 ± 112	468 ± 103	0.001	1233 ± 1327	255 ± 318	<0.001	12%	88%
van Thor, 2020 [59]	55	374 ± 124	422 ± 125	0.007	195 (96–1812)	154 (71–387)	0.075	63%	90%
Hoole, 2020 [60]	30	366 ± 107	440 ± 94	<0.001	442 (168–1607)	202 (105–447)	<0.001	20%	87%
Jansa, 2020 [53]	25	369 ± 116	54 ± 48 (absolute diff.) ^1^	<0.001	1107 ± 1458	−535 (949.4) (absolute diff.) ^1^	0.011	NR	NR
Gerges, 2021 [61]	45	NR	NR	<0.001	579 (182–1385)	198 (70–429)	<0.001	16%	91%
Darocha, 2021 [23]	70	365 ± 142	433 ± 120	<0.001	1307 (510–3294)	206 (83–531)	<0.001	20%	77%
Karyofyllis, 2022 [62]	15	NR	NR	-	925 ± 1238	231 ± 327	0.012	20%	80%
Wiedenroth, 2022 [63]	142	396 ± 109	440 ± 111	0.002	573 (132–1664)	109 (64–317)	<0.001	2.8%	91%
Darocha, 2022 [13]	156	341 ± 129	423 ± 136	<0.001	2275 (385–2675)	628 (85–533)	<0.001	2.94 ± 0.50	1.91 ± 0.64

absolute diff.: absolute difference; NR: not reported; NYHA-FC: New York Hearth Association functional class; 6MWD: 6-min walking distance; NT-proBNP: N-terminal fragment of pro-brain natriuretic peptide. ^1^ Absolute difference was calculated as the difference between values from the last BPA and before the first BPA.

## Data Availability

Not applicable.

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
