# Peer review of "Emerging Role of Balloon Pulmonary Angioplasty in Chronic Thromboembolic Pulmonary Disease—Insights from the 2022 ESC Guidelines"

_jcm, 2023, doi:10.3390/jcm12165336_

Round 1

Reviewer 1 Report

nice revew, well written and quite clear.

the future prospective are reported by authors.

I like the conclusion and the figures.

I should suggest to report specific drugs tested in CTEPH (bosentan...) and the medical indications according ESC/ERS GL (treprostinil, triple therapy...). a table could be added for drugs possibilities.

I would like to increase knowledge about CTPEH physiopathology.

discrete

Author Response

  • nice review, well written and quite clear.

Authors: Thank you very much for this comment.

  • the future prospective are reported by authors.

Authors: Thank you very much for this comment.

  • I like the conclusion and the figures.

Authors: Thank you very much for this comment.

  • I should suggest to report specific drugs tested in CTEPH (bosentan...) and the medical indications according ESC/ERS GL (treprostinil, triple therapy...). a table could be added for drugs possibilities.

Authors: Thank you very much for this comment.  The section on the use of pulmonary arterials targeted drugs in CTEPH has been modified according to your suggestions.   In addition, a table summarising the results of studies on these drugs in the CTEPH patient population has been added to the manuscript (Table 1).

Pretreatment with targeted pharmacotherapy

The ESC/ERS Guidelines provide class IIa recommendation for targeted medical therapy prior to BPA to manage the microvascular component of CTEPH[1]. Riociguat, the soluble guanylate cyclase (sGC) stimulator, has the highest class of recommendation, but other pulmonary arterial targeted drugs, such as phosphodiesterase type 5 inhibitors (PDE5i), endothelin receptor antagonists (ERA) or parenteral prostacyclin analogues may also be considered. The summary of pulmonary arterial targeted drugs tested in the CTEPH patient population is presented in Table 1. Moreover, a combination of sGC stimulator/PDE5i, ERA or pareneteral prostacyclin analogues may be used in selected group of patients[32,33] . Consequently, oral combination therapy including PDE5i and ERAs is a common practice in CTEPH patients with severe hemodynamic compromise [34].  The excellent results of the multimodal therapy with riociguat and BPA has been demonstrated in the study of 36 consecutive patients with inoperable CTEPH treated with riociguat for at least 3 months before BPA. Significant improvement in pulmonary hemodynamics and physical capacity has been observed for targeted medical treatment and subsequent BPA procedures yielded further benefits[35]. Moreover, pretreatment with riociguat improves hemodynamics and has been reported to reduce BPA-related complications. Thus, patients with a PVR above 4 Wood units should be treated with riociguat before starting interventional treatment [36].  To date, there are no clear recommendations regarding the termination or continuation of treatment after completion of BPA. The decision to continue or discontinue pulmonary arterial targeted treatment should be made individually, based on the patient’s symptoms, number of remaining lesions, hemodynamic parameters and previous response to treatment [29].

Agent

First author, year

Study type

Population [n]

Outcomes

Riociguat

Ghofrani[37]

2013

RCT

riociguat vs placebo

inoperable CTEPH or persistent CTEPH after PEA

[174 riociguat, 88 placebo]

6mWT distance increased by 46 m

PVR reduced by 246 dyn

mPAP decreased by 5 mmHg

NT-proBNP level decreased by 444 pg/ml

Jaïs[36]

2022

RCT

riociguat vs BPA

inoperable CTEPH and

PVR more than 320 dyn

[53 riociguat, 52 BPA]

PVR reduction and WHO FC improvement was more pronounced in BPA group

CI increased more in riociguat group

Sildenafil

Ghofrani[38]

2003

open-label pilot trial without control group

inoperable CTEPH and deterioration over a 3-month

[12 sildenafil]

PVR reduced by 574 dyn

6mWT distance increased by 54 m

mPAP decreased by 7.7 mmHg

Reichenberger[39]

2007

open-label trial without control group

inoperable CTEPH

[104 sildenafil]

6mWT distance increased by 51 m

PVR reduced by 104 dyn

Suntharalingam[40] 2008

RCT

sildenafil vs placebo

inoperable CTEPH or persistent CTEPH after PEA

[9 sildenafil, 10 placebo]

WHO FC improved in 44% cases

more than one

PVR decreased by 197 dyn

Bosentan

Hoeper[41]

2005

open-label pilot trial without control group

inoperable CTEPH or persistent CTEPH after PEA

[19 bosentan]

PVR reduced by 313 dyn

6mWT distance increased by 73 m

mPAP decreased by 6 mmHg

NT-proBNP level decreased by 716 pg/ml

Bonderman[42]

2005

case series

inoperable CTEPH

[16 bosentan]

WHO FC improved by one in 69% cases

6mWT distance increased by 92 m

NT-proBNP level decreased by 1610 pg/ml

Jaïs[43]

2008

RCT

bosentan vs placebo

inoperable CTEPH or persistent CTEPH after PEA

[77 bosentan, 80 placebo]

PVR decreased by 193 dyn

no significant differences in WHO FC

Macitentan

Ghofrani[33]

2017

RCT

macitentan vs placebo

inoperable CTEPH

[40 macitentan, 40 placebo]

PVR decreased by 16%

NT-proBNP level decreased by 1610 pg/ml

6mWT distance increased by 34 m

Treprostinil (s.c.)

Skoro-Sajer[44]

2007

prospective uncontrolled

observational cohort study

inoperable CTEPH or persistent CTEPH after PEA

[25 treprostinil]

6mWT distance increased by 105 m

WHO FC improved in 52% cases

PVR decreased by 116 dyn

Sadushi-Kolici[32]

2019

RCT

high-dose vs low-dose treprostinil

inoperable CTEPH or persistent CTEPH after PEA

[45 high-dose, 46 low-dose]

6mWT distance increased by 53 m

WHO FC improved in 51% cases

NT-proBNP level decreased by 157 pg/ml

PVR decreased by 214 dyn

Epoprostenol (i.v.)

Scelsi[45]

2004

retrospective study

inoperable CTEPH

[11 epoprostenol]

WHO FC improved in 55% cases

6mWT distance increased by 99 m

clinical signs of right heart failure resolute in 45% cases

Cabrol[46]

2007

retrospective study

inoperable CTEPH or persistent CTEPH after PEA

[27 epoprostenol]

WHO FC improved in 39% cases

6mWT distance increased by 66 m

mPAP decreased by 4 mmHg

Table 1. The summary of pharmacological therapies in chronic thromboembolic pulmonary hypertension.

CI – cardiac index, CTEPH – chronic thrombo-embolic pulmonary hypertension, mPAP – mean pulmonary artery pressure, NT-proBNP – N-terminal prohormone of brain natriuretic peptide, PEA – pulmonary endarterectomy, PVR – pulmonary vascular resistance, RCT – randomized controlled trial, WHO FC– World Health Organization functional class, 6mWT – six minutes walking test

  • I would like to increase knowledge about CTPEH physiopathology.

Authors: Thank you very much for this comment. The section “CTEPH pathophysiology” has been added to the manuscript.

Epidemiology and pathophysiology of CTEPH

The incidence of CTEPH is estimated to be around 3% in the first two years after a symptomatic episode of acute pulmonary embolism[3]. However, up to 25% of patients with CTEPH diagnosis have no history of a thromboembolic episode [4]. The pathophysiological basis of the disease lies in the incomplete resolution of thromboembolic lesions after at least 3 months of therapeutic anticoagulation. A characteristic feature is the fibrous remodelling of thrombi in the pulmonary arteries, resulting in permanent mechanical obstruction. It is unclear what determines incomplete thrombus resolution. Several factors are suggested, such as autoimmune diseases, haematological disorders or multiple comorbidities that impair the lytic system[5]. However, PH in this setting may not only be a consequence of pulmonary vascular bed obstruction by organized fibrotic clots. It may also be associated with the microvasculopathy, as a result of an increased blood flow in thrombus-free arteries. The morphology of the changes in the damaged microvasculature is similar to that observed in pulmonary arterial hypertension (PAH).  Altogether, all of the above processes result in an increase in pulmonary vascular resistance (PVR), which in turn leads to right heart failure (RHF) and death, if left untreated [6].

Reviewer 2 Report

The manuscript emphasizes BPA role in CTEPH, is clear, relevant for the field and presented in a well-structured manner. 

Minor changes could be considered.

1.      BPA Technical aspects:

            It is of great importance to note that BPA must be performed with biplane  cineangiography as it is a complex procedure requiring a detailed pulmonary vascular  anatomy.

            Contrast allergy and renal dysfunction should be taken in consideration.

A RHC is necessary before each BPA session to evaluate haemodynamic parameters.

2.     Systematic data regarding CTEPD  prognostic impact and therapeutic management are lacking. According to recent ESC Consensus on BPA in CTEPD, ‘both PEA or BPA should be considered in selected symptomatic patients with CTEPD without PH. For the identification of these patients, results of echocardiography, lung function testing, BNP/NT-proBNP, chest radiography, and cardiopulmonary exercise testing are considered in a multiparametric approach’.

European Heart Journal (2023) 00, 1–13,  ttps://doi.org/10.1093/eurheartj/ehad413

3.      The minimum haemodynamic goal of BPA is a final mPAP<30mmHg.

European Heart Journal (2023) 00, 1–13,  ttps://doi.org/10.1093/eurheartj/ehad413

4.      Concerning the medical treatment after BPA, patients’ symptoms, the number of remaining lesions and haemodynamics as also the response to previous medical treatments should be taken account.

5.      Line 115: in symptomatic patients,   (BPA should be performed only in symptomatic patients)

6.      Line 47: Right Heart failure instead of HF.

Kind Regards

Author Response

The manuscript emphasizes BPA role in CTEPH, is clear, relevant for the field and presented in a well-structured manner. 

Authors: Thank you very much for this comment.

Minor changes could be considered.

  1. BPA Technical aspects:

            It is of great importance to note that BPA must be performed with biplane  cineangiography as it is a complex procedure requiring a detailed pulmonary vascular  anatomy.

            Contrast allergy and renal dysfunction should be taken in consideration.

A RHC is necessary before each BPA session to evaluate haemodynamic parameters.

Authors: Thank you very much for this comment. The 'Technical Aspects' section has been rewritten and all of the above comments have been included.

Technical aspects

As balloon pulmonary angioplasty is the complex procedure requiring a detailed pulmonary vascular anatomy, it should be performed in catheter laboratory with biplane cineangiography [29,47]. In the process of preprocedural planning, contrast allergy, renal and/or thyroid dysfunction should be taken into consideration and handled according to general guidelines recommendations [29]. The procedure is performed in patients under local anesthesia. Vascular access is achieved via the femoral or jugular vein. It is necessary to perform right heart catheterization (RHC) before each BPA session to assess a number of hemodynamic parameters.

  1.    Systematic data regarding CTEPD  prognostic impact and therapeutic management are lacking. According to recent ESC Consensus on BPA in CTEPD, ‘both PEA or BPA should be considered in selected symptomatic patients with CTEPD without PH. For the identification of these patients, results of echocardiography, lung function testing, BNP/NT-proBNP, chest radiography, and cardiopulmonary exercise testing are considered in a multiparametric approach’.

European Heart Journal (2023) 00, 1–13,  ttps://doi.org/10.1093/eurheartj/ehad413

Authors: Thank you very much for this comment. The 'Patient management and selection' section has been rewritten and the above comment has been included.

Finally, ESC/ERS guidelines provide IIa-C recommendations on both PEA and BPA in the treatment of selected symptomatic patients suffering from CTEPD without PH [1] .To date, only small series have been published in this area[26] supporting this treatment option. However, the recently published ESC Consensus on BPA in CTEPD recommended both BPA and PEA to be considered in selected symptomatic patients with CTEPD without PH.  The multiparametric approach, consisting of cardiopulmonary exercise testing, echocardiography, lung function testing, BNP/NT-proBNP and chest radiography is thought to be useful in identifaction of these patients[ consensus EHJ] .

  1. The minimum haemodynamic goal of BPA is a final mPAP<30mmHg.

European Heart Journal (2023) 00, 1–13,  ttps://doi.org/10.1093/eurheartj/ehad413

Authors: Thank you very much for this comment. The 'Technical Aspects' section has been modified and above comment has been included.

The key factor for achieving a good hemodynamic effect is maximally extensive revascularization. Thus, angioplasty should be gradually pursued within all lung segments [49,50]. In general, according to ESC Consensus on BPA, the minimum hemodynamic goal of BPA is a final mPAP below 30mmHg.  However, it should be individualized, especially  in patients with comorbidities [29].

  1. Concerning the medical treatment after BPA, patients’ symptoms, the number of remaining lesions and haemodynamics as also the response to previous medical treatments should be taken account.

Authors: Thank you very much for this comment. The “Pretreatment with targeted pharmacotherapy” section has been modified and above comment has been included.

To date, there are no clear recommendations regarding the termination or continuation of treatment after completion of BPA. The decision to continue or discontinue pulmonary arterial targeted treatment should be made individually, based on the patient’s symptoms, number of remaining lesions, hemodynamic parameters and previous response to treatment [29].

  1. Line 115: in symptomatic patients,   (BPA should be performed only in symptomatic patients)

Authors: Thank you very much for this comment. The text has been redrafted.

  1. Line 47: Right Heart failure instead of HF.

Authors: Thank you very much for this comment. The text has been redrafted.

Round 2

Reviewer 1 Report

the authors responded to all my suggestions, improving the quality of the paper

well